# Knowledge, Behaviour and Attitudes Related to Sun Exposure in Sportspeople: A Systematic Review

**DOI:** 10.3390/ijerph191610175

**Published:** 2022-08-17

**Authors:** Jonatan Fernandez-Ruiz, Trinidad Montero-Vilchez, Agustin Buendia-Eisman, Salvador Arias-Santiago

**Affiliations:** 1Dermatology Department, Faculty of Medicine, University of Granada, 18071 Granada, Spain; 2Department of Dermatology, Hospital Universitario Virgen de las Nieves, 18012 Granada, Spain; 3Instituto de Investigación Biosanitaria de Granada (ibs.GRANADA), 18012 Granada, Spain

**Keywords:** exercise, sport, skin cancer, sun exposure, sun protection, sun-related behaviour, ultraviolet radiation

## Abstract

People who practice outdoor sports have an increased risk of skin cancer as they are exposed to high doses of ultraviolet (UV) radiation. Recent studies have shown that in many athletes, sun protection behaviours are inadequate, with the risk that this entails. The aim of this review is to collect the information published to date about the knowledge, attitudes and habits of athletes in relation to sun exposure and its risks. A systematic review was conducted using PubMed and Embase with the search algorithm “(skin cancer OR melanoma) AND (exercise OR sport OR athletes)”. All studies analysing the knowledge, attitudes and habits of photoprotection in athletes were included. A total of 2,365 publications were found, of which 23 were selected, including a total of 10,445 sportspeople. The majority of participants declared their voluntary intention to tan and stated that the sun made them feel better, although they also showed concern about possible damage associated with UV radiation. In most studies, less than half of the participants made adequate use of photoprotective measures. In general, most athletes had a high level of knowledge regarding the risk of skin cancer associated with sun exposure. In conclusion, most athletes are aware of the risks associated with UV radiation but do not make adequate use of photoprotective cream. New training programs on photoprotection could help improve athletes’ photoprotective behaviour, reducing the incidence of skin cancer and precancerous lesions in this population.

## 1. Introduction

Skin cancer is the most common type of cancer, which is divided into melanoma and nonmelanoma skin cancer (NMSC) [1]. Melanomas only represent 1% of skin cancer but are associated with higher mortality, with ultraviolet (UV) radiation being the most important modifiable risk factor for developing this tumour. The type of sun exposure influences the clinicopathological melanoma type. Intermittent sun exposure or sunburn during childhood and adolescence predisposes individuals to have superficial spreading melanoma, while chronic sun exposure predisposes individuals to develop lentigo maligna melanoma. Nodular melanomas have been connected to both sporadic and chronic sun exposure [2,3]. NMSC includes basal cell carcinoma (BCC) and squamous cell carcinoma (SCC), as well as other less common types of skin cancer. BCC is the most common malign tumour in humans, representing up to 60% of all skin tumours and is associated with intermittent sun exposure throughout life [2,4]. SCC is related to cumulative sun exposure over time [4].

Increased life expectancy, improvements in early detection as well as new habits derived from social trends regarding sun exposure, especially artificial tanning through UV ray equipment, have led to an increase in the incidence of skin cancer, significantly increasing the risk of melanoma [5]. Many studies have shown the important role physical exercise plays in cancer prevention, not only reducing its incidence, recurrence or mortality but also improving symptoms derived from the disease itself, as well as from the oncological therapies that may cause multiple side effects in patients [6,7,8].

Natural Killer (NK) cell activation through epigenetic mechanisms, decreased resistance to insulin, enhanced DNA repair mechanisms and increased antioxidant capacity are some of the molecular mechanisms that explain the biological plausibility of physical exercise in cancer prevention [9,10,11]. In some in vitro studies carried out on samples of mice in order to evaluate the protective effects of exercise on skin cancer induced by UV-B rays, downregulation was observed in the insulin-like growth factor type I (IGF-1) pathway through p53 activation [12,13]. However, research carried out on humans regarding physical exercise as a protective factor against cancers such as melanoma has not been able to provide such conclusive results, probably due to the confounding effect of sun exposure that very often accompanies physical exercise [14]. In fact, it has been observed that individuals who perform sport outdoors have a greater risk of skin cancer and a higher prevalence of pigmented lesions in areas exposed to the sun [15,16]. Similarly, it has been observed that the absence of good photoprotective practices in sportspeople is associated with a greater risk of skin cancer [17], which suggests that these measures may help to reduce the risk of skin cancer in some sportspeople.

The incidence of skin cancer continues to increase [3], while a large part of the population does not adopt adequate protective behaviours when they are outdoors, which may have a major impact on groups with greater exposure, with the added risk that UV ray intensity may be higher in many of the places where these sports are practised [16].

The aim of this review is to collect the information published to date about the knowledge, habits and attitudes of sportspeople regarding sun exposure and its risks, which could lead to considering new hypotheses regarding the need to implement prevention and healthcare programmes with the aim of reducing the incidence of skin cancer in this group. The research question was: What is the knowledge, behaviour and attitude related to sun exposure in sportspeople?

## 2. Materials and Methods

A systematic review was conducted using PubMed and Embase with the search algorithm “(skin cancer OR melanoma) AND (exercise OR sport OR athletes)” from the inception of the databases to 27 May 2022, following PRISMA guidelines (Appendix A). Published studies carried out on humans containing any of the following variables were included: photoprotective habits, attitudes to sun exposure and knowledge about sun exposure and its risks. All studies performed on sportspeople of any discipline were included, without excluding any age range or previous skin diseases. Articles not including sportspeople or not including information about knowledge, behaviours or attitudes regarding photoprotection were excluded. Non-epidemiological studies, including abstract conferences, case reports, case series and reviews, were also excluded. Articles that were not written in English, Spanish or Portuguese were excluded.

Two researchers (JFR and SAS) independently reviewed the titles and abstracts of the articles obtained in the first search to assess relevant studies. The full texts of all articles meeting the inclusion criteria were reviewed, and their bibliographic references were checked for additional sources. The articles considered relevant by both researchers were included in the analysis. Disagreements about the inclusion or exclusion of articles were subjected to discussion until a consensus was reached. If no consensus was reached, a resolution was achieved by discussion with a third researcher (ABE).

To assess the risk of bias, we followed the criteria provided by the Agency for Healthcare Research and Quality (AHRQ) for cross-sectional studies [18,19]. We chose this tool because it (1) includes transparency and reproducibility of judgments, (2) separates risk of bias from other aspects, such as applicability and precision, (3) evaluates the risk of bias per outcome and (4) focuses on reporting quality based only on study design or numerical quality scores [18,19].

The variables assessed were: type of sport, number of participants, author, country, age, sex, phototype, attitudes, behaviours, habits and knowledge related to sun exposure.

## 3. Results

The literature search identified 2365 references. After reviewing the title and abstract, 486 articles underwent full-text review, and 23 of them were selected for inclusion [20,21,22,23,24,25,26,27,28,29,30,31,32,33,34,35,36,37,38,39,40,41,42]. The flow chart of the articles selected is represented in Figure 1.

### 3.1. Socio-Demographic Characteristics

Socio-demographic characteristics of the participants included in each study are shown in Table 1. The total number of sportspeople evaluated was 10,445. Most of the studies included between 100 and 200 participants [20,27,28,32,37,41]. From all the studies found, 11 were published in Spain [24,25,26,27,29,30,32,33,34,39] and 5 in the United States [23,35,38,40,42]. Most studies evaluated runners or athletes [20,21,22,23,31,35,36,39,40,42]. Other sportspeople included were cyclists [30,38], golfers [29,37], volleyball players, skaters, hockey players, surfers, kitesurfers, windsurfers, sailors or swimmers. All the studies collected the data through items or questions adapted from the surveys used in previous research. In most cases, questionnaires adapted from validated surveys about behaviour and photoprotective habits and knowledge about sun exposure were used. In the majority of the studies, the questionnaire was completed in person and self-administered, except in three studies where it was completed online [31,38,42] and two studies where information was gathered through direct interviews [29,41].

In general, there were more male participants than females. Regarding age distribution, most studies included people between 20 and 50. Other studies evaluated younger people, with an average age around 14–15 [23,32,36]. The articles about golfers included older participants, with an age range between 60 and 65 [29,37].

Most studies used the Fitzpatrick scale [43] to collect data on participants’ phototype. Phototype III was the most common, at over 70%. Five studies showed phototype IV or higher as the most common [25,27,33,37,39].

### 3.2. Attitudes Related to Sun Exposure

In general, the sportspeople surveyed in the different studies admitted that they wanted to tan voluntarily while doing sport.

McCarthy et al. observed that more than half the sportspeople admitted wanting to tan voluntarily, and 40% said that they would not go to the doctor in the event of observing a new mole or a change in its appearance [37]. In six other studies, almost half the participants declared that the sun made them feel better and that they wanted to be tanned [24,30,33,34,39,41]. In the article by De Gálvez et al. on sportspeople who had previously undergone a solid organ and bone marrow transplant, almost half the participants showed a desire to tan voluntarily, while almost a third stated that they were not worried about possible sunspots or photoaging caused by the sun [28]. In another study, including young athletes, almost 20% agreed that sun exposure could improve appearance generally, and almost 30% stated that sun exposure helped them look healthier [35]. In addition, 83% of those surveyed stated that UV ray equipment could improve their health [35]. In the same line, in the study by Laffargue et al., 14% of the athletes associated tanning with health and 30% with beauty [36]. In the same study, 39% stated that it was worth getting sunburned to get a good tan [36]. In contrast to these studies, in the article by Petty et al. on a sample of almost 1000 cyclists, only 6% of participants stated their intention to tan voluntarily [38].

In five studies, more than 80% of sportspeople said that they were concerned about developing skin cancer due to the sun and that they were worried about being sunburned and blemishes or wrinkles appearing [24,30,33,34,39]. In addition, the same proportion of people surveyed preferred to stay in the shade at midday and stated that it was worth using sun protection. Only in one study, the number of subjects who preferred to be in the shade at midday to avoid direct sun exposure did not exceed 50% [24]. However, in all of them, more than half of the sportspeople had been sunburned at least once in the season before [24,30,33,34,39]. Furthermore, in six studies [24,26,27,36,38,42], more than 70% of the athletes had been sunburned during training in the year before and only in four studies [21,28,29,31] was the incidence of sunburn less than a third. The studies published by Wysong et al. and De Castro-Maqueda et al. showed the highest rates of sunburn at 84% and 84.7%, respectively [24,42].

### 3.3. Behaviour and Habits Related to Sun Exposure

In studies by McCarthy et al. and Gutiérrez-Manzanedo et al., a photoprotective cream use of 85% and 84%, respectively, was observed [34,37], while in another three studies, the proportion of sportspeople using sunscreen was almost 80% [24,26,38]. However, in some studies, more than 50% of those surveyed used a sun protection factor of 30 or less and mentioned not reapplying the cream after an hour or two. The results of four articles revealed sunscreen use of almost or slightly higher than 60% [28,29,33,39], and in the remaining cases, less than half of the athletes stated that they use sunscreen as a photoprotective measure. In the latter articles, half of the participants said they did not reapply photoprotective cream after one or two hours or use a sun protection factor higher than 30. In some cases, the rate of participants who always used sunscreen was around 15% [21,31,32,36,41,42]. Table 2 shows the frequency of photoprotective cream use, as well as other photoprotective habits by the participants in each study.

Other photoprotective measures used were sunglasses, caps or hats, long-sleeved clothing, seeking shade to do sport or training in time zones where the UV index was not high. In four studies, sunglasses were the main photoprotective measure, with a use higher than 80% in two studies [33,34] and at 66% [31] and 30% [39] in the other two studies, although in three of them, more than 40% of athletes did not use sunglasses and in only one study was the proportion of athletes using it equal to sunglasses use [34].

In three studies, the use of a cap or hat was less than 20% [27,31,35], in four, it did not exceed 40% [25,32,36,39], and in two studies, half of the athletes mentioned not using them [24,33]. Their use was greater than 70% in only two articles [34,37]. In the study by Del Boz et al., it was the most common measure, being used by 66.4% of participants [29], like in the study by Doncel-Molinero et al., where 95% of cyclists mentioned using it continuously [30].

The use of long-sleeved clothing was the least frequently used measure in almost all studies, with a proportion lower than 10% in the majority of them. Only in four studies was long-sleeved clothing used by more than 50% of those surveyed [25,27,32,34], being the most commonly used photoprotective method in the study on skaters and elite watersports athletes [25,32].

Regarding doing sport at times of low UV index, in six studies [24,29,30,32,34,35], more than 70% of sportspeople mentioned not avoiding the times of greatest risk and in three studies [27,33,36,39,41], only about half the athletes avoided training at midday. Only one study was found with a large proportion of sportspeople (81%) who avoided exposure during this time [31]. By contrast, in most studies, less than half of the sportspeople declared that they stay in the shade as much as possible. Only one study was found where the percentage of athletes who tried to seek shade was over 70% [25].

### 3.4. Knowledge about Sun Exposure and Associated Risks

Most sportspeople had a clear understanding of the risks associated with sun exposure. In most studies, more than 90% of the athletes showed knowledge of accelerated ageing related to UV radiation and the increased risk of skin cancer, and more than 80% knew that UV ray equipment significantly increases the risk of skin cancer [24,29,30,32,34,35]. In the study carried out on Irish golfers, around 70% of participants understood the risk associated with the presence of multiple nevi and the increased risk of melanoma with inappropriate sun exposure [37]. However, almost half of the golfers studied did not know that fair skin increased the risk of skin cancer. The highest error rates were related to knowledge about the need to sunbathe in order to satisfy vitamin D requirements, exposure to UV radiation despite being in the shade and the greater or lesser protection that clothing of different colours may entail [37]. Accordingly, in most studies, more than 80% of sportspeople had the incorrect idea that they needed an hour of sun exposure daily to satisfy vitamin D requirements and thought that the best measure for decreasing the risk of skin cancer was photoprotective cream. Furthermore, more than 80% stated that light clothing protects better than black against UV radiation, and almost half of those surveyed thought that staying in the shade was not enough to avoid skin cancer associated with UV radiation [24,28,30,33,34,39].

De Castro-Maqueda et al. observed that the subgroup of adolescent beach volleyball players had significantly lower scores than the group of university students in the section on knowledge about photoprotection [27]. In this study, 35% of the adolescents did not know that photoprotective cream prevents photoaging and protects them from the effect of solar radiation. In addition, 30% thought that using sunscreen prevented all risks, and almost 20% did not know that sun exposure is the main risk factor for skin cancer. Moreover, 25% thought that once tanned, sun protection is not necessary [27]. In the study by Hobbs et al., more than half of the athletes were not aware of the need to apply photoprotection 15–30 min before sun exposure, and 65% did not know about the need to reapply sunscreen every hour [35]. In addition, 80% did not know that a longer period of sun exposure carries a higher risk of skin cancer [35]. Likewise, in another study, only 66% of participants were aware of the risks of sun exposure regarding skin cancer [41].

Although, in some studies, more than 90% of sportspeople were aware of the risk of melanoma with sun exposure, less than 40% knew about the risk of basal cell carcinoma or squamous cell carcinoma [31].

### 3.5. Risk of Bias in the Studies Included

All the studies included were cross-sectional that assesses knowledge, attitudes and behaviours through self-administered questionnaires, so the risk of bias was high. Most of the studies included had a moderate risk of bias, but 13% of them (3/23) had a high risk of bias. All articles apply inclusion/exclusion criteria uniformly to all comparison groups. The design and analysis control account for important confounding and modifying variables in only 13% (3/23) of the studies. None of the researchers ruled out any impact from a concurrent intervention or an unintended exposure that might bias results, but it was because there was not any intervention (cross-sectional studies). Attrition or blinding was not a problem in any of the studies because there was no follow-up. Outcome measures and confounding variables were reliable and valid in all the studies included. The potential outcomes were not prespecified in only 17.4% (4/23) of the studies. This information is recorded in Appendix A.

## 4. Discussion

Sportspeople are moderately well informed about photoprotection habits and the risks associated with sun exposure. However, most people who do outdoor sports do not show good photoprotection habits and associate tanning with a healthy lifestyle.

Thus far, several reviews have been carried out on the incidence and prevalence of skin cancer and behaviour towards sun exposure in different groups, such as workers [44] or in the military [45]. However, to date, there has been no review of habits and behaviours, attitudes and knowledge related to sun exposure and photoprotection in sportspeople.

The beneficial health effects of sport are widely recognised [6,7,8]. However, many studies have shown that outdoor activities increase the risk of skin cancer and precancerous lesions. Hållmarker et al. observed an increased risk of skin cancer in a cohort of over 180,000 skiers compared to another cohort from the general Swiss population [46]. In addition, they saw a lower incidence of skin cancer in skiers who finished the race in less time. In another study on a sample of 210 runners, a significantly higher number of atypical nevi and solar lentigines were observed compared to a control group matched for age and sex, with even greater differences in subjects with a higher training intensity [47]. Jardine et al. observed a significant increase in the risk of having been sunburned in the last year in sportspeople, with a stronger association in those with a higher number of training hours [48].

Most studies evaluating the habits, attitudes and behaviour about sun exposure and photoprotection in sportspeople who do sport outdoors have been done in Spain mainly on groups of runners or athletes, collecting the information through validated and self-administered health questionnaires on photoprotection and sun exposure habits based on previous research on prevention behaviours in sportspeople. The number of articles has also increased over the years, likely in relation to the increase in skin cancer by the end of the first decade of this century [5].

In general, most sportspeople were aware of the risks of sun exposure and the effects of UV radiation related to photoaging and the increased risk of skin cancer. However, there is a discordant pattern between the level of knowledge and the level of use of photoprotective cream and other photoprotective measures, as more than 80% of sportspeople were aware of the associated risks but less than half used photoprotective cream [24,39,42]. In addition, in many studies where adequate levels of sunscreen use were observed as a preventative measure against sunburn, the majority of participants stated that they used sunscreen with a low sun protection factor, and more than 80% of participants did not reapply the protective cream after one or two hours of use [27,33,41].

Among the measures used for sun protection were photoprotectors, sunglasses, long-sleeved clothing, caps or helmets. In some studies, using a cap or helmet was the main measure, ahead of the use of sunscreen or sunglasses [29,30,32]. Such a varied use of different measures between studies may be explained by some measures being associated to a greater or lesser extent with one type of sport, according to the activity the athletes perform when competing [49]. Thus, for example, in the study by Doncel-Molinero et al. on a group of cyclists, the most-used measure was the helmet, possibly more associated with preventing injury than photoprotection [30]. Similarly, in articles on golfers, using caps not only protects against UV radiation but also makes it easier to follow the path of the ball by protecting the eyesight from the sun’s rays [29,37]. By contrast, in some sports, caps or sunglasses may be uncomfortable or result in extra difficulties for competing.

Most sportspeople declared that they were worried about possible sunburn, photoaging or the appearance of blemishes, as well as the risk of skin cancer due to sun exposure. However, almost half of those surveyed in the different studies stated that they wanted to get a suntan and that they feel better in the sun, associating sun exposure with a good health index. In this way, this generalised thinking about the need for sun exposure to get good vitamin D rates stands out. It has been observed that sun exposure increases vitamin D but also T-T dimer values in urine, which reflects DNA damage caused by UV radiation [50]. After observing the low percentage of sportspeople who use photoprotective cream and other measures, it would be interesting to educate the population about safer ways to obtain vitamin D and encourage the use of different photoprotective measures in people who do sport.

The results observed in the different studies also reflect the need for education in aspects such as the type and colour of clothing that should be used, the need to use sunscreen even if you stay in the shade or under the sunshade, and above all to emphasise that the most important measure for decreasing the risk of skin cancer is to avoid sun exposure during times when the UV index is at its highest.

It is important to also mention that sports that involve increased perspiration require more photoprotective measures as sweat can influence the hydration of the corneal layer of the skin, causing less reflection and dispersion of UV light and increasing the risk of sunburn [51].

Promising measures to improve photoprotective habits are trainers’ recommendations [42], raising awareness among athletes’ parents about the need for good photoprotective habits [36] and implementing educational programmes on photoprotection habits in young athletes, trainers and coaches [52,53,54]. It would be interesting to invest in urban design standards, with spaces protected from the sun in parks and sports fields to facilitate doing sport outdoors in the shade [55].

Among the limitations found in the review is the healthy volunteer bias since the subjects most concerned about their health will be those who are most represented in these studies. Despite this, the sunburn rate was high (over 50%) in most cases, so it would not be surprising to find much higher rates of sunburn in other sample types. We also included articles that collected information about sportspeople with previous skin disease. Further research about sun habits should collect information about this issue as people with and without previous cutaneous pathologies could have different sun-exposure habits.

In addition, most of the subjects included in the different studies were men, with a significantly lower number of women. It was also found that the female sex was associated with a significant increase in photoprotective measures. Although women showed a greater tendency to associate tanning with better health status, they also showed higher use of photoprotective cream generally [31,35]. These observations agree with the results of other studies that found an association between the female sex and a greater desire to get a tan but also better protective habits [56,57]. Therefore, it would not be surprising if including a greater number of women in these studies would have provided different results regarding habits and attitudes to sun exposure.

Another limitation was the lack of information regarding some of the questions raised in the review. Not all the articles contained data on habits, attitudes and knowledge. By contrast, the use of the same validated questionnaires in most of the studies is noteworthy and has facilitated information synthesis and data comparability between studies. Moreover, all the studies included were cross-sectional that recorded information using self-administered questionnaires, so the level of evidence is limited.

## 5. Conclusions

Most sportspeople are aware of the risks associated with photo exposure but generally show inadequate photoprotection habits and attitudes and a high incidence of sunburn. It is necessary to implement educational health programmes to improve the behaviours and attitudes of sportspeople about sun exposure. Improvement behaviour regarding photoprotection in those collectives would help to decrease the incidence of precancerous lesions and skin cancer.

## Figures and Tables

**Figure 1 ijerph-19-10175-f001:**
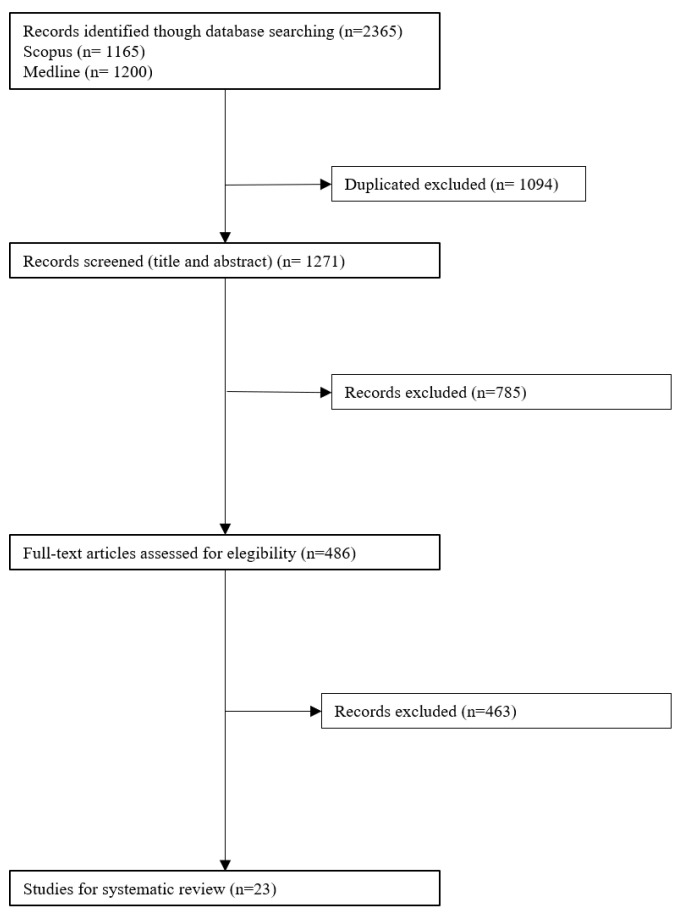
PRISMA flow diagram of the search strategy.

**Table 1 ijerph-19-10175-t001:** Demographic characteristics of the participants.

References	Geographical Area	Sport	Method of Data Collection	Number of Participants in the Study	Gender (% Males)	Age	Distribution of Participants’ Phototypes (%)				
Bakos et al. [20]	Brazil	Athletics	Self-administered questionnaire	115	53	F: 23.5 M: 21.7	Light: 59.1		Dark: 40.9		
Buljan et al. [21]	Croatia	Athletics	Self-administered questionnaire	95	68	NS	NS				
Christoph et al. [22]	Switzerland	Athletics	Self-administered questionnaire	970	47	41	I: 17.9	II: 41.1	III: 26.8	IV: 14.2	
Cohen et al. [23]	United States	Athletics	Self-administered questionnaire	612	52.4	15.2 (1 SD)	Men I–II: 35 Women I–II: 27				
De Castro-Maqueda et al. [24]	Spain	Kitesurfing	Self-administered questionnaire	72	100	25.9 (6.4 SD)	I–II: 69.4				
De Castro-Maqueda et al. [25,26]	Spain	Surfing, windsurfing and sailing	Self-administered questionnaire	240	70	22 (5.8 SD)	Men ≤ III: 64 Women ≤ III: 84				
De Castro-Maqueda et al. [27]	Spain	Beach handball players	Self-administered questionnaire	124	55 ^a^ 85 ^b^	22 ^a^ 12 ^b^	≤III ^a^: 34 ≤III ^b^: 42				
De Gálvez et al. [28]	Spain	Athletics, swimming and cycling	Self-administered questionnaire	166	71	48 (range 6–78)	I: 3.1	II: 43.6	III: 38.7	IV: 18	
Del Boz et al. [29]	Spain	Golf	Direct interview	342	56	51.9 (14.1)	I–II: 51.5		III–IV: 48.5		
Doncel Molinero et al. [30]	Spain	Cycling	Online questionnaire	1018	88	41.8 (11 SD)	I: 4.6	II: 33	III: 51.4	IV: 10.9	
Duarte et al. [31]	Portugal	Athletics	Online questionnaire	2445	75	25–44 range	≤III: 57				
Fernández-Morano et al. [32]	Spain	Skate	Self-administered questionnaire	122	82	14.3 (2.3 SD)	I: 5.9	II: 23.5	III: 36.3	IV: 34.3	
García-Malinis et al. [33]	Spain	Long-distance race	Self-administered questionnaire	657	72	39.7 (7.9 SD)	I-II: 21.4		III-IV: 78.6		
Gutiérrez-Manzanedo et al. [34]	Spain	Sailing	Self-administered questionnaire	60	80.4	40.6 (14.1 SD)	I: 17.9	II: 41.1	III: 26.8	IV: 14.3	
Hobbs et al. [35]	United States	Athletics	Self-administered questionnaire	393	55	20	NS				
Laffargue et al. [36]	Argentina	Athletics	Self-administered questionnaire	554	45	14.7	≤III: 65				
McCarthy et al. [37]	Ireland	Golf	Self-administered questionnaire	163	84	65	I: 4	II: 28	≥III: 68		
Petty et al. [38]	United States	Cycling	Online questionnaire	927	75	48 (11 SD)	≤III: 56				
Rivas-Ruiz et al. [39]	Spain	Long-distance race	Self-administered questionnaire	273	83	40.1 (9 SD)	I-II: 21.3		III–IV: 78.7	III–IV: 78.7	III–IV: 78.7
Tenforde et al. [40]	United States	Runners	Self-administered questionnaire	697	41.8	42.3 (13.1 SD)	NS				
Walker et al. [41]	New Zealand	Rugby, hockey and rowing	Direct interview	110	69	23.5 (3.1 SD)	II: 32		III: 38	IV: 16	V: 14
Wysong et al. [42]	United States	Athletics	Self-administered questionnaire	290	42	18–24 range	I–II: 34	III: 39	IV: 18	V–VI: 9	

^a^: subgroup of university students; ^b^: subgroup of adolescents, F: females; M: males, NS: not specified; SD: standard deviation.

**Table 2 ijerph-19-10175-t002:** Frequency of use of photoprotective measures by sportspeople.

References	Photoprotective Measures (%)					
	Photoprotective Cream	Sunglasses	Cap/Hat	Long Sleeves	Shade	Training at Times of Lower UV Index
Bakos et al. [20]	*NS*					
Buljan et al. [21]	*a:* 3 *f*: 27 *r*: 50 *n*: 20	NS	NS	NS	NS	NS
Christoph et al. [22]	<50	NS	NS	NS	NS	NS
Cohen et al. [23]	40.3	22.9	27.2	24.4	NS	NS
De Castro-Maqueda et al. [25,26]	79.2	62.5	50	20.8	30.6	6.9
De Castro-Maqueda et al. [24]	*n*: 22.5	44.3	*n*: 31.7 *r*: 30.4	80	75	NS
De Castro-Maqueda et al. [27]	17.8 ^a^ 31.3 ^b^	27.4 ^a^ 4.2 ^b^	NS	5.5 ^a^ 10.4 ^b^	11 ^a^ 20.8 ^b^	NS
De Gálvez et al. [28]	69	NS	NS	NS	NS	NS
Del Boz et al. [29]	66	56.2	69.1	8.3	45.1	30.4
Doncel Molinero et al. [30]	39.2	92.8	95.5	6.2	23.1	50
Duarte et al. [31]	12	30	17	4	NS	81
Fernández-Morano et al. [32]	18.7	20	33.3	65.9	33.3	23.3
García-Malinis et al. [33]	62	74	52	7.4	40	53
Gutiérrez-Manzanedo et al. [34]	83.9	85.7	75	48.2	43.6	28.6
Hobbs et al. [35]	23	NS	12	12	NS	14.9
Laffargue et al. [36]	5.2	NS	37	NS	NS	NS
McCarthy et al. [37]	85	NS	74	NS	NS	NS
Petty et al. [38]	75	NS	NS	NS	NS	NS
Rivas-Ruiz et al. [39]	58.2	66.4	33.6	10.9	50.6	56.3
Tenforde et al. [40]	42	45	NS	7	40	43
Walker et al. [41]	45 (*a*: 9)	NS	NS	NS	NS	NS
Wysong et al. [42]	40	NS	NS	NS	NS	NS

*a*: always; *f*: frequently; *r*: rarely; *n*: never, ^a^: subgroup of university students; ^b^: subgroup of adolescents, NS: not specified.

## Data Availability

The data presented in this study are available on request from the corresponding author.

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
