# Peer review of "Knowledge, Behaviour and Attitudes Related to Sun Exposure in Sportspeople: A Systematic Review"

_ijerph, 2022, doi:10.3390/ijerph191610175_

Round 1

Reviewer 1 Report

Introduction: define NK, IGF-1

Methods:

Time frame for literature inclusion not specified (from the inception of database)?

The authors did not filter articles based on English. How did they handle articles published in languages they do not understand?

Sport persons with existing/previous pathologies were included. This deserve a separate discussion.

What do the authors mean “Not epidemiological studies”?

The research question should be mentioned in the last paragraph of introduction rather than methods.

It was unclear how bias assessment performed for each study. What is the guidelines used?

Results:

Please recheck the number in PRISMA flow chart. (record screened: 1,2071)

The geographical distribution of the studies found was surprisingly limited (USA and Spain). I repeated the search using PubMed and found several potential studies outside these countries:

https://pubmed.ncbi.nlm.nih.gov/27232425/

https://pubmed.ncbi.nlm.nih.gov/26999653/

https://pubmed.ncbi.nlm.nih.gov/30698149/

Did the authors limit their search to these two countries only?

Table 1: some of the age data are given as range, not average. Please also provide the SD of the mean as well.

No table to summarise the knowledge and attitude studies.

The results of bias risk assessment should be tabulated.

Are there any studies to compare the KAP of UV protection between the sportmen and the average public?

Discussion:  Limitation of the review itself should be discussed.

Conclusion: Should be written in paragraph.

Author Response

Introduction: define NK, IGF-1

These acronyms have been defined: Natural Killer (NK), insulin-like growth factor type I (IGF-1)

Methods:

Time frame for literature inclusion not specified (from the inception of database)?

We meant from database creation. We have checked this issue.

The authors did not filter articles based on English. How did they handle articles published in languages they do not understand?

Thank you for your comment. As answered in the previous letter, all articles reviewed in English, Portuguese or Spanish, languages understood by the authors. As suggested, we have included it in the inclusion criteria: Articles that were not written in English, Spanish or Portuguese were excluded

Sport persons with existing/previous pathologies were included. This deserve a separate discussion.

We have explained this issue and have included more information in this discussion: We also included articles that collected information about sportspeople with previous skin disease. Further research about sun-habits should collect information about this issue as people with and without previous cutaneous pathologies could be different sun-exposure habits.

What do the authors mean “Not epidemiological studies”?

We meant case reports, case series and reviews. We have checked this in the article

The research question should be mentioned in the last paragraph of introduction rather than methods.

Thank you for the comment. The question has been included where suggested.

It was unclear how bias assessment performed for each study. What is the guidelines used?

As there is no validated tool to assess the risk of bias for systematic reviews including cross-sectional studies, we followed the criteria provided by Visawanathan et al[18] for cross-sectional studies. We have also included a supplementary table (Table S2).

Results:

Please recheck the number in PRISMA flow chart. (record screened: 1,2071)

Thank you for the comment. It has been checked

The geographical distribution of the studies found was surprisingly limited (USA and Spain). I repeated the search using PubMed and found several potential studies outside these countries:

https://pubmed.ncbi.nlm.nih.gov/27232425/

https://pubmed.ncbi.nlm.nih.gov/26999653/

https://pubmed.ncbi.nlm.nih.gov/30698149/

Did the authors limit their search to these two countries only?

We did not limit the research to any country. The first research mentioned (https://pubmed.ncbi.nlm.nih.gov/27232425/) was also found in using the search algorithm in our systematic review but it was not included because it assessed the prevalence of different types od skin cancer but they did not included information about knowledge, behavior or attitude. The other researches (https://pubmed.ncbi.nlm.nih.gov/26999653/, https://pubmed.ncbi.nlm.nih.gov/30698149/) are included in our systematic review: Christoph et al.[21] and Duarte et al.[30]

Table 1: some of the age data are given as range, not average. Please also provide the SD of the mean as well.

Data is provided as each article gave then. If possible mean (SD) was used but if it was not provided, data used in the original research was reported.

No table to summarise the knowledge and attitude studies.

Thank you for the comment. We only provided this information in the text because It is difficult to standardise this section because of the way in which each article presents this data.

The results of bias risk assessment should be tabulated.

We have included a new table (table s2) with tabulated risk of bias.

Are there any studies to compare the KAP of UV protection between the sportmen and the average public?

Up to our knowledge there aren’t any. Please if you know some, it would be interesting to include It in the discussion. If any, it is an interesting topic to do further research.

Discussion:  Limitation of the review itself should be discussed.

Limitations have been widely discussed.

 Conclusion: Should be written in paragraph.

Conclusion have been written in paragraph now. It was itemizing because it was a suggestion from the previous revision.

Reviewer 2 Report

The authors had answered point by point all the questions

The manuscript has been significantly improved

Author Response

Thank your very much for your review and suggestions

Round 2

Reviewer 1 Report

Minor edits are required but the editor can check on my behalf: 

Line 66: change outside to outdoor. 

Line 81: from inception "of databases", delete (since database origin)

Table 1: Change average age to age. Indicate the values individually it is mean or range. If SD values are available from the article, the authors should indicate them in the table.

Author Response

Minor edits are required but the editor can check on my behalf: 

Thank you for the comments

Line 66: change outside to outdoor. 

It has been modified

Line 81: from inception "of databases", delete (since database origin)

It has been changed

Table 1: Change average age to age. Indicate the values individually it is mean or range. If SD values are available from the article, the authors should indicate them in the table.

It has been indicated if it is a mean or a range individually. SDs have been added if available

This manuscript is a resubmission of an earlier submission. The following is a list of the peer review reports and author responses from that submission.

Round 1

Reviewer 1 Report

The review is interesting and will help sports decision makers however, these are my observations

1. the authors should check the fonts for uniformity 

2. they should check the gramma

3. the subsections should be numbered

4. i suggest itemizing and improving the conclusions

Reviewer 2 Report

Some critical elements of systematic review are lacking: 

1. Only one database is used for the literature search. At least 2 are recommended.

2. There is no rationale given for limiting the search results the latest 10 years.

3. The searched keywords might not be comprehensive enough. Synonyms like "melanoma" and "athletes" are not included.

4. How did the authors overcome the language barrier if an article is not published in English?

5. There is no element of critical appraisal in this systematic review. It is not formally done and the limitations/quality of the studies included are not discussed.

6. Discussion: First paragraph should be removed.

7. I recommended the authors to download the PRISMA checklist and cross-check their manuscript with the list. 

8. The PRISMA flow chart in the manuscript does not follow the standard of PRISMA. Please check again.

Reviewer 3 Report

The manuscript is a systematic review on photoprotection and sun exposure in people who practice their sport activity outdoors. It is an interesting article which provides important information about the attitudes, knowledge and behaviors related to sun exposure in sportpeople. The methodology is described in detail, initially identifying 158 studies in PubMed, finally comparing 19 selected articles and the results are presented in the form of clear tables. The final conclusion is clear: most athletes are aware of sun-related damage and are concerned about the risks associated with UV radiation but, do not take the necessary photoprotection measures.

I recommend the article for publication after some minor revisions

The following changes are suggested:

Page 1, line 38: Nodular melanomas have been connected to both sporadic and chronic sun exposure

 Add reference

Page 4, line 96: After  reviewing the references of all the articles, two more publications were included

¿after how many articles?. It is not clear where the 2 studies added come from . It is also not clear in the flow diagram of the Fig 1 (PRISMA).

Page 4 line 104

¿Had been published ?

Page 5, line 135: Regarding age distribution, in seven studies the age was between 40 and 50 years  old [22–24,28,29,36,41]

Item number 22 is not in that age range. In the table (page 6) the average age is 51.

Page 9, line 208

creams or sunglasses?

Page 10, line 247

It seems that being in the shade increases the risk of skin cancer, as if being in the shade is negative. It would be better to express that being in the shade is not enough to avoid skin cáncer

Page 11, line 297.:The amount of skin cancer has doubled and even tripled by the end of the first decade of this century

Add reference